# Plant miRNAs for Improved Gene Regulation in a Wide Range of Human Cancers

**DOI:** 10.3390/cimb47010042

**Published:** 2025-01-09

**Authors:** Maksym Zoziuk, Vittorio Colizzi, Pavlo Krysenko, Maurizio Mattei, Roberta Bernardini, Fabio Massimo Zanzotto, Stefano Marini, Dmitri Koroliouk

**Affiliations:** 1Centro Interdipartimentale di Medicina Comparata, Tecniche Alternative ed Acquacoltura, Interdepartmental Center for Comparative Medicine, Alternative Techniques, and Aquaculture, University of Rome Tor Vergata, Via Montpellier 1, 00133 Rome, Italy; zzkmsy01@uniroma2.it (M.Z.); mattei@uniroma2.it (M.M.); 2UNESCO Chair in Interdisciplinary Biotechnology, University of Rome Tor Vergata, Via della Ricerca Scientifica 1, 00173 Rome, Italy; colizzi@uniroma2.it; 3Institute of Telecommunications and Global Information Space of the National Academy of Sciences of Ukraine, Chokolivskiy bulv. 13, 03186 Kyiv, Ukraine; p.krysenko@gmail.com; 4Department of Biology, University of Rome Tor Vergata, Via della Ricerca Scientifica 1, 00173 Rome, Italy; 5Faculty of Medicine, Department of Clinical Sciences and Translational Medicine, University of Rome Tor Vergata, Via Montpellier 1, 00133 Roma, Italy; roberta.bernardini@uniroma2.it (R.B.); stefano.marini@uniroma2.it (S.M.); 6Department of Business Engineering “Mario Lucertini”, University of Rome Tor Vergata, Via del Politecnico, 1, 00133 Rome, Italy; znzfms00@uniroma2.it; 7Department of Microelectronics, Faculty of Electronics, National Technical University of Ukraine “Igor Sikorsky Kyiv Polytechnic Institute”, Beresteiska Ave. 37, 03056 Kyiv, Ukraine; 8Institute of Mathematics of the National Academy of Sciences of Ukraine, 3, Tereschenkivska St., 01004 Kyiv, Ukraine

**Keywords:** miRNA expression, cancer treatment, statistical complementarity, plant miRNAs

## Abstract

Determining the relationships between miRNA expression, target genes, and cancer development is critical to cancer research. The possibility of correlating miRNA expression with plant or artificial ones provides prerequisites for cancer treatment. Based on the broad database of human miRNA expression for all cancer types, we grade human miRNAs by their expression level. The identified deficient miRNAs are compared with their target genes for coincidences in their expression directions. The replacement of human miRNAs is proposed to be implemented, using plant miRNAs closest to the human-deficient ones. Such plant substitutes are identified by analyzing the average complementarity of all human under-expressed miRNAs. It was established that the number of downregulated miRNAs is almost 2.5 times greater than that of upregulated miRNAs. There is no significant correlation between the expression of miRNA and genes, implying many other expression regulation mechanisms exist. Working on the organization of experimental verification of the obtained statistical studies, we present significant regularities that provide grounds for considering some plant microRNAs as possible means of compensating for insufficient expression of regulatory microRNAs in humans and animals in a wide range of oncological diseases.

## 1. Introduction

miRNAs are short RNAs that play an important role in post-transcriptional gene regulation. Their expressions can change constantly under the influence of various factors, such as, for example, the presence of a disease [1,2,3] or microbial infections [4]. One idea is to regulate the progression of the disease through the regulation of gene and miRNA expressions. There are more than 3000 different miRNAs in the human body. Their expressions change dynamically depending on various factors and signals that affect the cells. Controlling the regulation of specific miRNAs is complex and does not give hope for their successful use because one miRNA can have more than a hundred target genes.

Recent studies have shown that specific miRNAs can be biomarkers for cancer diagnosis and prognosis [5]. For example, three miRNAs predicts the risk of colorectal cancer predicts the risk of colorectal cancer that helps predict the risk of colorectal cancer, which has been identified and significantly improve the prognostic performance of baseline models [6]. Other studies have shown that increased expression of miRNA-146a in chronic hepatitis can impair T cell activity, which contributes to the progression of liver cancer [7]. In addition, miRNAs can be used as therapeutic targets. For example, therapies aimed at inhibiting oncogenic miRNAs or inducing tumor suppressors have shown efficacy in cancer treatment [8].

Plant miRNAs are often more structurally stable, target mRNA with near-perfect complementarity, and primarily regulate genes involved in stress responses and development. In contrast, human miRNAs typically bind their targets partially, influencing a broader array of processes such as immune responses, neuroplasticity, and cancer progression. Studies like [9,10,11,12,13] highlight these distinctions, underscoring the specialized roles of miRNAs across species and their relevance in health and disease contexts.

Some studies have explored the potential of plant-derived microRNAs (miRNAs) as therapeutic agents for human diseases, including cancer. Research indicates that dietary intake of plant miRNAs can influence human health by modulating gene expression. For instance, a review published in *Frontiers in Genetics* discusses how plant miRNAs can regulate human gene expression, suggesting their potential as therapeutic agents [14]. Additionally, a study in *Scientific Reports* demonstrated that plant miRNAs can regulate the expression of key human cancer-related genes in vitro, highlighting their potential role in cancer therapy [10]. Furthermore, research in *Frontiers in Genetics* indicates that plant miRNAs can reduce cancer cell proliferation by targeting specific genes, suggesting their potential as anti-cancer agents [15].

Considering the evolution of microRNA in all living organisms, another issue is that many miRNAs may have common functions. For plants, more than 80–90% similarity is required for mutual replacement of miRNAs, while 50% can be enough for animal organisms [15,16]. A study published in eLife indicates that while miRNA targets are often recognized through pairing between the miRNA seed region and complementary sites within target mRNAs, not all of these canonical sites are equally effective. Both computational and in vivo UV-crosslinking approaches suggest that many mRNAs are targeted through non-canonical interactions. However, the study also notes that recently reported non-canonical sites do not mediate repression despite binding the miRNA, indicating that the vast majority of functional sites are canonical [17]. Additionally, research published in Nucleic Acids Research discusses how most miRNAs are only partially complementary to their target sequences, and full or even majority complementarity is not necessary for effective gene silencing [18]. In this sense, it is necessary to analyze the complementarity of specific miRNAs with all others for the complete set of human miRNAs. Also, an important aspect is the use of plant miRNAs to regulate the expression of human miRNAs, because they are more accessible, compared to the creation of artificial ones or the extraction of animal ones.

Using these premises, it can be concluded that by establishing the appropriate interdependencies between human and plant miRNAs, as well as target genes, it is possible to identify plant miRNAs that can be used to replace human ones and, thus, correct the development of cancer or treat the organism. Also, the use of these correlations allows for the identification of important genes and human genes that need to be regulated in other ways and allows for the establishment of new hypotheses and the establishment of new directions of scientific research. We propose to use a statistical approach to classify miRNAs and their target genes according to their experimental characteristics. It was observed that the number of strongly under-expressed miRNAs significantly exceeds the number of overexpressed ones. However, no overall correlation was found between the expression levels of target genes and miRNAs. Despite this, specific correlations were identified for certain miRNAs and their target genes. Additionally, the potential for utilizing plant miRNAs to regulate and compensate for human miRNAs, along with the selection methods for this purpose, was explored.

## 2. Materials and Methods

The database [19] was used as a basis, in which the expression of miRNAs in various types of cancer and in various conditions are presented. The dbDEMC 3.0 database contains 158,196 records of miRNA expression in human cancer under various experimental conditions (see link for details). All data were used, and the possibility of establishing any correlations between all parameters in the database was studied—primarily the correlation between miRNA sequence and expression level. The data types are as follows: cancer type (most existing types), cancer subtype, cell type, experiment type (cancer vs. normal; high grade vs. low grade; subtype1 vs. subtype2; blood, etc.), logFC, average expression, statistical values, status (direction of expression—up/down). Information on miRNA sequences was taken from the miRBase database [20]. The database [21] of gene expression in various cancers and the database of target genes and miRNAs [22] were used to calculate statistical correlations. The OncoDB database contains 414,934 records for 20 cancer types with information on gene expression in the presence of these diseases. Data on logarithms of change in expression depending on the cancer type and gene were used. The miRTarBase database contains 1,048,575 records on the existence of relationships between miRNAs and target genes, which have been established experimentally. Information about these relationships was used directly.

For calculation of the main statistic parameter div, the following formula was used:(1)div=(numofdown−numofup)(numofdown+numofup)
where $num_{of_{down}}$—the number of experiments in which this miRNA is downregulated, and $num_{of_{up}}$—the number of experiments in which this miRNA is upregulated.

A fully connected artificial neural network based on tensor flow with three layers, with 35 internal neurons, was used. The input data were experimental conditions type and miRNA sequences. Preprocessing was performed using character-by-character encoding for miRNA, one hot for experimental conditions type, and numerical encoding for expression. We performed training and testing. The division into bulk and test datasets was performed with a 70/30 ratio. Additional information about pre-miRNA was also added, which did not give significant additional accuracy (+3–4%).

Figure 1 shows a block diagram of calculations to establish possible correlations. The dbDEMC 3.0 database is used to classify expressions with magnitude. Target genes are obtained from miRTarBase and their expressions from OncoDB. Sorting by the magnitude of expression by genes and miRNAs and by Formula (1). Comparison with plant miRNAs based on local and global complementarity (plant miRNA sequences are obtained from the PMRD database) [23]. Sorting by average complementarities between human and plant miRNAs and establishing useful plant miRNAs for regulating human expression in cancer.

For all statistical calculations, analysis, sorting, and displaying results, we used software developed by our team using popular Python libraries on PyCharm IDE 2022.1.

## 3. Results

### 3.1. Statistical Analysis of miRNA Expression

To establish dependence, an artificial neural network was used to predict miRNA expression based on various parameters—miRNA sequence, experimental conditions, type of cancer, etc. It was established that the accuracy of predicting the direction of expression (“up” or “down”) does not exceed 75% (±5 percent for various input parameters), which means that the dependence is present, but not complete (that is, there is information that is not taken into account, for example, genetic differences of individuals, and so on) [5].

This 25% of information can be partially clarified. The use of artificial intelligence is not effective in determining the exact source expression using only sequence information and experimental conditions. This means that the information that determines the expression of this miRNA does not depend only on the sequence. Other information can be the type of cancer, the type of cell, biological/medical indicators at the time of expression control, etc. Thus, only approximate information about expression can be used to analyze the dependencies between miRNA and cancer. We use a statistical approach to determine these dependencies and determine the possibility of establishing a correlation between genes, miRNAs, and the corresponding expressions. The machine learning method allows us to establish the presence of a correlation, and the statistical approach allows us to sort and establish a distribution according to quantitative characteristics—the number of experimental data and the number of experiments in which the direction/strength of expression is different (1). A statistical approach helps establish dependencies that can be used to identify a specific set of human miRNAs that tend to be up/down expressed.

As can be seen from Formula (1), the dependence of miRNA expression in the presence of a disease depends on the number of experiments in which it was measured and on the direction of expression (up or down). After that, it is possible to sort all metrics by div indicator and by the number of experiments. The miRNAs with the highest number of experiments, the largest and the smallest div should be examined in more detail and used to find target genes as well as plant substitutes for these miRNAs—Figure 2.

It was established that the number of downregulated miRNAs is almost 2.5 times greater than that of upregulated miRNAs. Given that the number of over-expressing and under-expressing miRNAs is nearly the same the quantitative characteristics of div and the number of experiments (reliability of mean expression) are not considered.

### 3.2. Dependence of miRNA Expression Statistics on Gene Expression Statistics

Each miRNA can have more than a hundred target genes. This means that by analyzing their statistical expressions, it is possible to assert the up/down expression of a specific set of genes. By analyzing the database [21] of gene expression in various cancers and the database of target genes and miRNAs [22], it is possible to compare the effect of miRNAs on target genes. Using the div parameter, but instead of the number of experiments, substitute the number of subject miRNAs (those targeting this gene/genes) and sorting by div and by the number of these miRNAs for each gene, we will establish a set of genes that are dependent on miRNA expression—Figure 3.

It was established that there is no significant correlation between the expression of miRNA and genes, which means that there are a large number of other mechanisms of expression regulation. By comparing these expressions and target genes, it is possible to establish where their behavior coincides and, thus, select only those miRNAs or their sets from those that have already been sorted beforehand. We are most interested in genes for which the *div* coefficient is positive. They represent greater practical significance than those genes for which *div* is negative because it is more difficult to change the concentration of upregulated miRNAs than the concentration of downregulated ones.

### 3.3. Establishing Plant miRNAs to Compensate for Human Ones

For each plant miRNA, one needs to calculate the average complementarity with all human miRNAs that have a clear trend of up/down expression and have similar target genes in the trend. Two numbers can be fixed for each plant miRNA—the average complementarity with upregulated and downregulated human miRNAs. After sorting all plant miRNAs according to these parameters, we will obtain a specific set that can be important in the compensation processes of downregulated miRNAs in the presence of cancer such as osa-miRf10192, ptc-miRf10867, and ath-miRf11181.

On the other hand, a large number of plant miRNAs are on average complementary to human upregulated ones such as osa-miRf11782-akr, gma-miR4396, and zma-miR156r. Therefore, sorting by both parameters is required (the highest average complementarity with downregulated and the lowest with upregulated miRNAs). Most plant miRNAs were found to have the same average complementarity as under-expressed and overexpressed human miRNAs. Only 5% of all plant miRNAs have a difference in mean complementarities greater than 10%. This suggests that it makes no sense to use specific plants at the expense of miRNA, since the miRNA contained in them will compensate for each other. On the other hand, the concentration of various miRNAs can vary greatly depending on growing conditions, land, etc.

## 4. Discussion

This work analyzes the process of selecting human-critical miRNAs that need to be investigated in more detail compared to others based on a statistical approach. By regulating a specific set of miRNAs, it is possible to regulate the course of the disease. Since in the presence of cancer, a large number of miRNAs are under-expressed or overexpressed, the question arises as to which miRNAs need to be regulated and how.

It is important to note that in the process of researching the accuracy of prediction using a universal approximator in the form of a neural network, we are unable to predict the expression of human miRNA depending on its nucleotide sequence [8]. More precisely, a prediction with an error is sufficient to call into question the possibility of its practical use. There is a high probability that the name miRNA is not only the sequence of miRNA but a set of other parameters that essentially characterize the expression of miRNA. What parameters can these be? Adjacent substances that are involved only in mechanisms specific to specific miRNAs. Output and input structures are characterized by specific measures and are not common to all. It is worth noting that approximately the same error is observed when analyzing the statistical data of miRNA expression. As the number of miRNA expression experiments increases, the mean expression value approaches zero. This may also be the reason for the study of not very informative miRNAs from the point of view of oncological mechanisms. It is necessary to pay attention to miRNAs for which, even with 40–50 experiments under different conditions in different types of tissues, and for different diseases, the direction of expression is observed to be stable.

Regulation of human miRNAs, which are in insufficient quantity in the body, is a more promising direction of development because methods of replacing human miRNAs with miRNAs that have arrived from the outside are a simpler means than reducing the concentration of miRNAs in an excessive amount. On the other hand, a decrease in miRNA concentrations can create a negative effect, because of a possible protective reaction of the body. Still, it can be in reverse, since in most cases the body cannot cure cancer on its own without external influence, and, therefore, the defensive system does not work, and a change in miRNA concentrations in the blood has to be more of a positive influence than a negative one. Individual miRNAs can be created artificially to obtain the highest average complementarity with human-critical miRNAs. Still, at the same time, the process of creating these miRNAs is costly when plant variants are more available.

It is worth remembering that the complementarity of 50%, which is sufficient for interchangeable activity, is determined to a different extent for different specific human miRNAs. Also, these values may vary depending on different conditions—depending on the organism, health status, etc. Further studies (for example, experimental observation for a large number of experiments to establish the average value of the required complementarity for each specific miRNA) are necessary to determine and test hypotheses.

## 5. Conclusions

A statistical approach to the analysis of miRNA expressions and the expressions of their target genes is proposed. It was established that the number of strongly under-expressed miRNAs is several times greater than overexpressed ones. On the other hand, no global correlation was established between the expression of target genes and miRNA. However, certain correlations are present for some miRNAs and target genes. The possibility of using and the method of selection of plant miRNAs for regulation and compensation of human ones is analyzed.

## Figures and Tables

**Figure 1 cimb-47-00042-f001:**
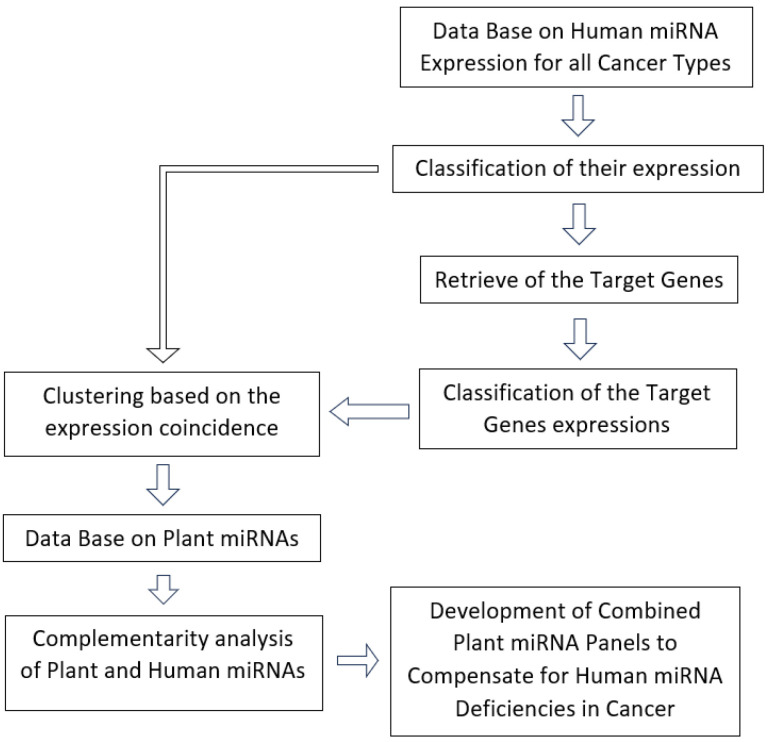
Flowchart of calculations for establishing dependencies and identifying important human and plant miRNAs, as well as genes.

**Figure 2 cimb-47-00042-f002:**
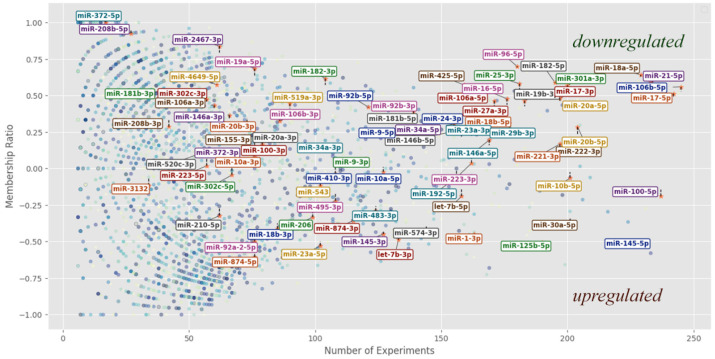
The graph of the distribution of unit miRNA depends on the number of experiments (X-axis) and membership ratio—“div” (Y-axis). The miRNA points for which the number of experiments does not exceed five are not shown (the number of all human 3171 miRNAs is reduced to 2607) [19].

**Figure 3 cimb-47-00042-f003:**
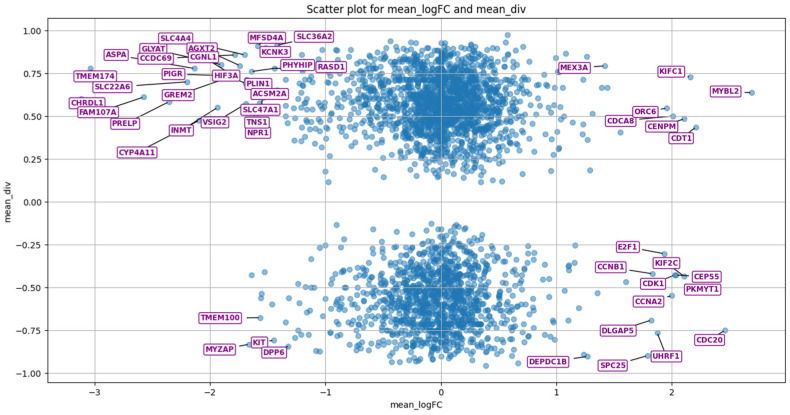
Location of genes according to mean logarithmic expression (mean_logFC—parameter was calculated as the mean value across all cancers for that individual gene) for all types of cancer and parameter (mean_div) based on miRNA expressions. The 50 most distant genes from the center are shown here as an example, although the number of interesting genes is not limited to this number. Each quarter represents a separate piece of information. The first is upregulated miRNAs (for these genes) and under-expressed genes. The second—downregulated miRNA, but overexpressed genes. The third is upregulated miRNA, but under-expressed genes. The fourth is upregulated miRNAs and under-expressed genes. Those genes (points) that are closer to zero by mean logFC have zero expression.

## Data Availability

Databases that were used in their original form are available at the links indicated in the work. Generated datasets used and analyzed during the current study are available from the corresponding author upon reasonable request.

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
