# Peer review of "Plant miRNAs for Improved Gene Regulation in a Wide Range of Human Cancers"

_cimb, 2025, doi:10.3390/cimb47010042_

Round 1

Reviewer 1 Report

Comments and Suggestions for Authors

This manuscript that entitled by “Plant miRNAs for improvement gene regulation in a wide range of human cancers”, conducted a research study by doing a statistical approach to the classification of plant and human miRNAs according to their experimental characteristics. They used known databases of miRNA and gene expressions in the presence of cancer. The findings stated that the number of miRNAs that are downregulated is almost 2.5 times greater than that of miRNAs that are upregulated. Also, there is no global correlation established between the expression of target genes and miRNA. The manuscript is generally well-addressed and well-cited; however, I have some comments/suggestions.

Line 25: The abstract is too long. Please rewrite to be more focus and specific.

Line33: Upon journal guidelines, the abstract should be a single paragraph and should follow the style of structured abstracts, but without headings: like Background, methods,.... as shown in your abstract. Please revise to follow the journal guidelines.

Line 47: upon database used as a references in the study, they showed data for mice and rat only like reference number 7. Please be focused and specific.

Line 48: I suggest to replace one of those related or add " Plant miRNAs" to be more specific to the article

Line 49: Please note that there are only 4 references used for the literature of this article. Please add more resources and rewrite to be more informative.

Line 62: Please add more explanation for your aim of work and why did you need to do that work, with more use of recent literature.

Line 68: Please add more evidences that plant miRNAs can be used to for human purposes.

Line 70: At the end of introduction please,  briefly mention and highlight the main conclusions or your findings.

Line 71: Please rewrite the materials and methods with more information and details about the database that you are using in your study.

Line 73: you mentioned reference number (9), however the reference number (8) didn't mention before. Please revise.

Line 79: please mention all statistical software that was used for database analysis and software was used for sorting and displaying your results. (name, version, year of production and source of this software)

Line 147: Although this very low (5%) difference between both plant and human miRNAs, do you think that plant miRNAs would work by the same mechanism of actions and regulation when used for human diseases like cancer? Please verify and add that point to the discussion.

Line 185: I suggest rewriting it again by a formal, simple and more informative way with a concise and precise description of the experimental results.

Line 199: Please mention the source of internal funds or even the location or institute  that approved to conduct this research study using their infrastructure or lab supplies.

Line 205: Please revise these information with that mentioned at the funding section on line 199.

References # 2, 4, 5 and 6 are incomplete. Please revise by following the journal guidelines that’s included: Abbreviated Journal Name Year, Volume, page range

Comments on the Quality of English Language

Major editing of English language required.

Author Response

-----------2-------------
Remarks to the authors:

Line 25: The abstract is too long. Please rewrite to be more focus and specific.
Changed. 

Line33: Upon journal guidelines, the abstract should be a single paragraph and should follow the style of structured abstracts, but without headings: like Background, methods,.... as shown in your abstract. Please revise to follow the journal guidelines.
Changed. 

Line 47: upon database used as a references in the study, they showed data for mice and rat only like reference number 7. Please be focused and specific.
Databases presented consist human data - miRNAs, genes. 

Line 48: I suggest to replace one of those related or add " Plant miRNAs" to be more specific to the article
Sorry, not very clear. 

Line 49: Please note that there are only 4 references used for the literature of this article. Please add more resources and rewrite to be more informative.
Added more references. 

Line 62: Please add more explanation for your aim of work and why did you need to do that work, with more use of recent literature.
Added more references and explanaition. 

Line 68: Please add more evidences that plant miRNAs can be used to for human purposes.
Added more references and explanaition. 

Line 70: At the end of introduction please,  briefly mention and highlight the main conclusions or your findings.
Added paragraph.

Line 71: Please rewrite the materials and methods with more information and details about the database that you are using in your study.
Added information.

Line 73: you mentioned reference number (9), however the reference number (8) didn't mention before. Please revise.
Changed.

Line 79: please mention all statistical software that was used for database analysis and software was used for sorting and displaying your results. (name, version, year of production and source of this software)
Added sentence. 

Line 147: Although this very low (5%) difference between both plant and human miRNAs, do you think that plant miRNAs would work by the same mechanism of actions and regulation when used for human diseases like cancer? Please verify and add that point to the discussion.
Added.

Line 185: I suggest rewriting it again by a formal, simple and more informative way with a concise and precise description of the experimental results.
Changed.

Line 199: Please mention the source of internal funds or even the location or institute  that approved to conduct this research study using their infrastructure or lab supplies.
No finding was done. 

Line 205: Please revise these information with that mentioned at the funding section on line 199.
No finding was done. 

References # 2, 4, 5 and 6 are incomplete. Please revise by following the journal guidelines that’s included: Abbreviated Journal Name Year, Volume, page range
Done.

Reviewer 2 Report

Comments and Suggestions for Authors

Zoziuk et al. investigate the relationship between miRNA expression, gene expression, and cancer development, with a focus on underexpressed human miRNAs in various cancer types. They identify target genes whose expression patterns align with those of deficient miRNAs and propose compensating for these deficiencies using plant miRNAs that exhibit the highest complementarity to their human counterparts. Their findings highlight that downregulated miRNAs significantly outnumber upregulated ones but show no strong correlation between miRNA and gene expression, suggesting the involvement of additional regulatory mechanisms.

I think the paper needs to be improved before publication, especially in providing more details regarding the background and the experiments. Below, I provide several remarks to guide the authors in addressing potential areas for enhancement.

Remarks to the authors:

1) The Introduction provides a general overview of miRNAs and their regulatory roles but lacks depth in several key areas. To strengthen this section, consider expanding on the background regarding the relationship between miRNAs and human cancers, as this would provide a clearer context for the study's significance. Additionally, it would be valuable to elaborate on the differences between human and plant miRNAs, particularly with references to existing work, such as that by Diez-Sainz, Pastrello, Miskiewicz, Samad, etc. (DOIs: 10.1007/s13105-024-01023-0, 10.1038/srep32773, 10.1155/2017/6783010, 10.1093/advances/nmaa095, 10.3390/molecules23061367), which could help clarify the rationale for exploring plant miRNAs as regulators in human systems. Finally, the authors should explain how the concept of using plant miRNAs emerged in this study, as this would provide readers with a better understanding of the novelty and foundation of their approach.

2) "The database [7] was used", "The database [9]" - give the names of databases and explain what data (type and number of records) these databases contain, and which data you use in the study. The information provided is not complete. Besides, write what correlations you wanted to calculate (between which data).

3) "an artificial neural network was used to" - what kind of a network was it? There are many types. Please, give the details concerning the architecture and the AI predictive model as well as the reference to this model. Who and how trained the network? What was the training and test data? All these details should be given in the "Materials and methods" section.

4) "The use of artificial intelligence is 89 not effective in determining the exact source expression using only sequence information 90 and experimental conditions." - support this statement with a reference

5) Give details concerning experimental data, such as cardinality and contents of the datasets.

6) "Two numbers can be fixed for each plant miRNA - the average complementarity with up-regulated and down-regulated human miRNAs. After sorting all plant miRNAs according to these parameters" - where can I find the list of all these miRNAs with parameters? Supplementary information is needed here.

7) "This work analyzes the process of selecting human critical miRNAs that need to be investigated in more detail compared to others based on a statistical approach" - I suggest adding the figure with workflow to illustrate the designed study. It should improve the clarity of the paper.

Author Response

-----------1-------------
Remarks to the authors:
1) The Introduction provides a general overview of miRNAs and their regulatory roles but lacks
depth in several key areas. To strengthen this section, consider expanding on the background
regarding the relationship between miRNAs and human cancers, as this would provide a clearer
context for the study's significance. Additionally, it would be valuable to elaborate on the differences
between human and plant miRNAs, particularly with references to existing work, such as that by
Diez-Sainz, Pastrello, Miskiewicz, Samad, etc. (DOIs: 10.1007/s13105-024-01023-0, 10.1038/srep32773,
10.1155/2017/6783010, 10.1093/advances/nmaa095, 10.3390/molecules23061367), which could help
clarify the rationale for exploring plant miRNAs as regulators in human systems. Finally, the authors
should explain how the concept of using plant miRNAs emerged in this study, as this would provide
readers with a better understanding of the novelty and foundation of their approach.
Relevant paragraphs and links have been added to the Introduction.
2) "The database [7] was used", "The database [9]" - give the names of databases and explain what
data (type and number of records) these databases contain, and which data you use in the study. The
information provided is not complete. Besides, write what correlations you wanted to calculate
(between which data).
Relevant paragraphs and links have been added to the Materials and Methods.
3) "an artificial neural network was used to" - what kind of a network was it? There are many types.
Please, give the details concerning the architecture and the AI predictive model as well as the
reference to this model. Who and how trained the network? What was the training and test data? All
these details should be given in the "Materials and methods" section.
Relevant paragraphs have been added to the Materials and Methods.
4) "The use of artificial intelligence is not effective in determining the exact source expression using
only sequence information and experimental conditions." - support this statement with a reference
Relevant paragraphs have been added to the Results.
5) Give details concerning experimental data, such as cardinality and contents of the datasets.
Relevant paragraphs have been added to the Materials and Methods.
6) "Two numbers can be fixed for each plant miRNA - the average complementarity with up-
regulated and down-regulated human miRNAs. After sorting all plant miRNAs according to these
parameters" - where can I find the list of all these miRNAs with parameters? Supplementary
information is needed here.
Unfortunately, we are unable to add these materials because we reserve the right to use these data
for experimental testing.
7) "This work analyzes the process of selecting human critical miRNAs that need to be investigated
in more detail compared to others based on a statistical approach" - I suggest adding the figure with
workflow to illustrate the designed study. It should improve the clarity of the paper.
Added block-scheme into the work and accorded explanaition.

Round 2

Reviewer 1 Report

Comments and Suggestions for Authors

The manuscript is improved than before. Thank You!.  But, still have a little comments: 

Line 40: The introduction is too long. Please rewrite to be more specific and focused.

Line 77: Please verify what do you mean by "eLife". Is it a journal or web site or book or conference?

Line 109: Concerning the (see link for details), Please add the link or reference for it to the text to be easily accessible by readers.

Line 124: Please note that there is (1). Do you mean formula 1? Please add it to the text not as that location. Also, is there formula 2? If not, you don't need to mention the number for formula as it is the only one used at the study.

Line 134: That's looks like a figure legend while it’s a description or part within the text. Please rewrite this paragraph to be as a part of your materials and methods section. There is already a figure legend written at line 143.

Line 146: Please add more details about the PyCharm IDE as a source of software used in your study. Please keep it clear for other researchers about how to use it after you're used it.

Line 155: Please verify why you add reference number to your results?

Line 168: Please verify what do you mean by (1)? Is it formula one? Also, please review that mentioned in lines 137 and 171.

Comments on the Quality of English Language

Minor editing of English language required.

Author Response

-----------1-------------

Remarks to the authors:

1) The Introduction provides a general overview of miRNAs and their regulatory roles but lacks depth in several key areas. To strengthen this section, consider expanding on the background regarding the relationship between miRNAs and human cancers, as this would provide a clearer context for the study's significance. Additionally, it would be valuable to elaborate on the differences between human and plant miRNAs, particularly with references to existing work, such as that by Diez-Sainz, Pastrello, Miskiewicz, Samad, etc. (DOIs: 10.1007/s13105-024-01023-0, 10.1038/srep32773, 10.1155/2017/6783010, 10.1093/advances/nmaa095, 10.3390/molecules23061367), which could help clarify the rationale for exploring plant miRNAs as regulators in human systems. Finally, the authors should explain how the concept of using plant miRNAs emerged in this study, as this would provide readers with a better understanding of the novelty and foundation of their approach.

Comment: The relevant paragraphs and links have been added to the Introduction.

2) "The database [7] was used", "The database [9]" - give the names of databases and explain what data (type and number of records) these databases contain, and which data you use in the study. The information provided is not complete. Besides, write what correlations you wanted to calculate (between which data).

Comment: The relevant paragraphs and links have been added to the Materials and Methods.

3) "An artificial neural network was used to" - what kind of a network was it? There are many types. Please, give the details concerning the architecture and the AI predictive model as well as the reference to this model. Who and how trained the network? What was the training and test data? All these details should be given in the "Materials and methods" section.

Comment: The relevant paragraphs have been added to the Materials and Methods.

4) "The use of artificial intelligence is not effective in determining the exact source expression using only sequence information and experimental conditions." - support this statement with a reference

Comment: The relevant paragraphs have been added to the Results.

5) Give details concerning experimental data, such as cardinality and contents of the datasets.

Comment: The relevant paragraphs have been added to the Materials and Methods.

6) "Two numbers can be fixed for each plant miRNA - the average complementarity with up-regulated and down-regulated human miRNAs. After sorting all plant miRNAs according to these parameters" - where can I find the list of all these miRNAs with parameters? Supplementary information is needed here.

Comment: Unfortunately, we cannot add these materials because we reserve the right to use these data for experimental testing.

7) "This work analyzes the process of selecting human critical miRNAs that need to be investigated in more detail compared to others based on a statistical approach" - I suggest adding the figure with workflow to illustrate the designed study. It should improve the clarity of the paper.

Comment: Added block scheme into the work and accorded explanation.

Reviewer 2 Report

Comments and Suggestions for Authors

The authors have responded to all my comments and questions. I have no further questions regarding the study. However, the authors should standardize the formatting of references, as they are using at least three different styles of citation in the bibliography.

Author Response

-----------2-------------

Remarks to the authors:

Line 25: The abstract is too long. Please rewrite to be more focus and specific.

Comment: Changed.

Line33: Upon journal guidelines, the abstract should be a single paragraph and should follow the style of structured abstracts, but without headings: like Background, methods,.... as shown in your abstract. Please revise to follow the journal guidelines.

Comment: Changed.

Line 47: upon database used as referencesv in the study, they showed data for mice and rat only like reference number 7. Please be focused and specific.

Comment: The databases presented consist of human data: miRNAs, and genes.

Line 48: I suggest to replace one of those related or add " Plant miRNAs" to be more specific to the article

Comment: Sorry, it is not very clear.

Line 49: Please note that there are only 4 references used for the literature of this article. Please add more resources and rewrite them to be more informative.

Comment: Added more references.

Line 62: Please add more explanation for your aim of work and why did you need to do that work, with more use of recent literature.

Comment: Added more references and explanations.

Line 68: Please add more evidences that plant miRNAs can be used to for human purposes.

Comment: Added more references and explanations.

Line 70: At the end of introduction please,  briefly mention and highlight the main conclusions or your findings.

Comment: A paragraph was added.

Line 71: Please rewrite the materials and methods with more information and details about the database that you are using in your study.

Comment: Added more information.

Line 73: you mentioned reference number (9), however the reference number (8) didn't mention before. Please revise.

Comment: Changed.

Line 79: please mention all statistical software that was used for database analysis and software was used for sorting and displaying your results. (name, version, year of production and source of this software)

Comment: Added more information.

Line 147: Although this very low (5%) difference between both plant and human miRNAs, do you think that plant miRNAs would work by the same mechanism of actions and regulation when used for human diseases like cancer? Please verify and add that point to the discussion.

Comment: Added more information.

Line 185: I suggest rewriting it again by a formal, simple and more informative way with a concise and precise description of the experimental results.

Comment: Changed.

Line 199: Please mention the source of internal funds or even the location or institute  that approved to conduct this research study using their infrastructure or lab supplies.

Comment: No funding was used.

Line 205: Please revise this information with that mentioned at the funding section on line 199.

Comment: No funding was used.

References # 2, 4, 5 and 6 are incomplete. Please revise by following the journal guidelines that’s included: Abbreviated Journal Name Year, Volume, page range

Comment: Done.